# The In Vivo Toxicity and Antimicrobial Properties for Electrolyzed Oxidizing (EO) Water-Based Mouthwashes

**DOI:** 10.3390/ma13194299

**Published:** 2020-09-26

**Authors:** Yi-Ling Hsieh, Jiun-Cheng Yao, Sung-Chih Hsieh, Nai-Chia Teng, You-Tai Chu, Wen-Xin Yu, Chung-He Chen, Liang-Yu Chang, Ching-Shuan Huang, Tzu-Hsin Lee, Aivaras Kareiva, Jen-Chang Yang

**Affiliations:** 1Graduate Institute of Nanomedicine and Medical Engineering, College of Biomedical Engineering, Taipei Medical University, Taipei 110–52, Taiwan; r29948264@gmail.com (Y.-L.H.); jacky.yao55@gmail.com (J.-C.Y.); chudaev1977@ukr.net (Y.-T.C.); andy54861@gmail.com (W.-X.Y.); gpchaucer@gmail.com (C.-H.C.); alex627324@gmail.com (L.-Y.C.); tzu6415@hotmail.com (T.-H.L.); 2School of Dentistry, Taipei Medical University, Taipei 110–52, Taiwan; endo@tmu.edu.tw (S.-C.H.); tengnaichia@hotmail.com (N.-C.T.); jollyhuangtw12@gmail.com (C.-S.H.);; 3Institute of Chemistry, Vilnius University, Naugarduko 24, LT-03225 Vilnius, Lithuania; aivaras.kareiva@chgf.vu.lt; 4Research Center of Biomedical Device, Taipei Medical University, Taipei 110–52, Taiwan; 5International Ph.D. Program in Biomedical Engineering, College of Biomedical Engineering, Taipei Medical University, Taipei 110–52, Taiwan; 6Research Center of Digital Oral Science and Technology, Taipei Medical University, Taipei 110–52, Taiwan

**Keywords:** in vivo toxicity, electrolyzed oxidizing (EO) water, chlorhexidine (CHX) gluconate, zebrafish, *Streptococcus mutans* (*S. mutans*)

## Abstract

The objective of this study was to verify the feasibility of electrolyzed oxidizing (EO) water as a mouthwash through the evaluation of its in vivo toxicity by embryonic zebrafish and antimicrobial efficacy against *Streptococcus mutans* (*S. mutans*). Methodology: Each 1.5–3.0 g of sodium chloride (NaCl), sodium bromide (NaBr), or calcium chloride (CaCl_2_) were added into an electrolyzer with 300 mL of DD water to produce electrolyzed oxidizing (EO) water. A zebrafish embryo assay was used to evaluate acute toxicity of specimens. Antimicrobial property was conducted with 100 μL microbial count of 1 × 10^8^ cfu/mL *S. mutans* to blend with each 10 mL specimen of chlorhexidine (CHX) gluconate or hypochlorous acid (HOCl) for various time points. The concentration of viable microorganisms was assessed according to individually standardized inoculum by a plate-count method. Results: Among the EO water produced from NaCl, NaBr, and CaCl_2_, the EO water from NaCl showed a relatively low mortality rate of zebrafish embryos and was chosen for a detailed investigation. The mortality rates for the groups treated with EO water containing 0.0125% and 0.0250% HOCl were not statically different from those of a negative control, however the mortality rate was 66.7 ± 26.2% in 0.2% CHX gluconate for the same treatment time of 0.5 min. All of the HOCl or 2.0% CHX gluconate groups showed >99.9% antimicrobial effectiveness against *S. mutans;* while the 0.2% CHX gluconate group showed a bacterial reduction rate of 87.5% and 97.1% for treatment times of 0.5 min and 1.0 min, respectively. Conclusions: Except for the 0.2% CHX gluconate, all the HOCl specimens and 2.0% CHX gluconate revealed similar antimicrobial properties (>99.9%) against *S. mutans*. The EO water comprised of both 0.0125% and 0.0250% HOCl showed >99.9% antimicrobial efficacy but with little in vivo toxicity, illuminating the possibility as an alternative mouthwash for dental and oral care.

## 1. Introduction

Dental and oral health are of great importance for overall health and well-being. Upholding good oral hygiene is important to battle dental cavities and gum disease possibly linked to heart disease, cancer, and diabetes [1,2,3,4]. In addition to daily brushing and flossing, the use of mouthwashes in inaccessible areas such as proximal embrasures can significantly promote oral health [5]. There are two kinds of mouthwashes, namely, cosmetic and therapeutic [6]. Unlike cosmetic mouthwashes that are usually used for temporary bad breath control, therapeutic mouthwashes include antimicrobial agents for effective bacterial reduction and prevention of receding gums, gingivitis, dry mouth, and plaque buildup [7]. Typical therapeutic mouthwashes are comprised of active ingredients such as cetylpyridinium chloride for reducing bad breath [8], chlorhexidine (CHX) gluconate for plaque and gingivitis control [9], fluoride for decay prevention [10], and peroxide for tooth whitening [11].

Among the active ingredients for mouthwash, CHX gluconate is currently known as the gold standard as an antiplaque agent. However, CHX gluconate rinsing has been associated with some side effects such as stained teeth and tongue [12], human taste perception disturbance [13,14], and minor irritation with superficial desquamation of oral mucosa [15]; thus, considerably limiting a patient’s acceptance of CHX mouthwash [16].

Typically, electrolyzed oxidizing (EO) water is produced using a dilute sodium chloride aqueous solution in an electrolysis chamber with a permeation membrane between the anode and cathode [17]. Acidic electrolyzed oxidizing (EO) water with a low pH (2–3) but high oxidation reduction potential (ORP, >1000 mV) is produced in the anode [18]. EO water comprised of hypochlorous acid (HOCl), which is the same chemical generated by the human body’s immune cells to combat infections, usually possesses a surprising bactericidal effect [19]. Cellular death is attributed to the disruption of bacterial adenosine triphosphate (ATP) production by oxidative and fermentative pathways by HClO [20]. EO water is effective against a wide range of microorganisms and it is used for many applications in medical devices such as root canal irrigation [21], scar prevention [22], wound biofilms [23], and inflammatory skin disorders [24]. The antiplaque effect [25] and bacterial viability [26] of hypochlorous acid mouthwash have been investigated; while the effects of contact time and dose dependent toxicity study have been left neglected.

An ideal therapeutic mouthwash with both biocompatibility and antimicrobial properties is important for effective disinfection in dental care and oral hygiene. Our purpose was to verify the feasibility of electrolyzed oxidizing (EO) water as a mouthwash through the evaluation of its in vivo toxicity by embryonic zebrafish and antimicrobial efficacy and to compare it with CHX gluconate mouthwashes.

## 2. Materials and Methods

### 2.1. Production of Electrolyzed Oxidizing (EO) Waters

Three different salts of sodium chloride (NaCl) (BioShop Canada Inc., Burlington, ON, Canada), sodium bromide (NaBr) (AppliChem, Darmstadt, Germany), and calcium chloride CaCl_2_ (calcium chloride dihydrate), Granular, Baker Analyzed™ A.C.S. Reagent, (J.T. Baker™, Phillipsburg, NJ, USA) were purchased and used to prepared EO water without further purification. The TC X-7 fungicide generator (ZhongShan Tiancheng Electrical Appliances Co., Zhongshan, China) was utilized to produce EO water by mixing DD water with various salts and operated under 5 V and 5 W for 8 min. The pH value and the oxidation-reduction potential (ORP) of EO waters were measured by a pH/mV/CON/TDS/SAL/DO/°C multifunction water analysis meter (HD-PHC700S, Hondwen, Taipei, Taiwan) equipped with pH electrodes (TN-T651-GB, Hondwen, Taipei, Taiwan) and ORP sensor (ORP-CN GB07986J, Hondwen, Taipei, Taiwan). Then, the pH of EO water was attuned to between pH 5.5 and 6.5. The free chlorine or bromine concentration of the EO water was tested using a HI96771 chlorine photometer (Hanna Instruments, Woonsocket, RI, USA). The free chlorine or free bromine content of harvested EO water was in the range of 0.025–0.040%, first diluted to 0.018% for screening out a group with the least toxicity. Then, diluted EO waters with halogen contents of 0.0125% and 0.0250% were further tested for their in vivo toxicity and antimicrobial properties. The control groups of 0.2% and 2.0% CHX gluconate solution were prepared by mixing 5 mL of reagent grade of 20% CHX gluconate solution (Millipore Sigma, St. Louis, MO, USA) into 45 mL and 495 mL of DD water, respectively.

### 2.2. In Vivo Toxicity Assays

The animal study ethics approval (LAC-2020-0194) from the Taipei Medical University ethics committee was obtained, although no permit was required if the zebrafish embryos used were less than 5 days old according to the European Union, Directive 2010/63/EU (revised from Directive 86/609/EEC) [27]. Two hundred and seventy fertilized wild-type zebrafish (Danio rerio) eggs, 1 h post fertilization (1 hpf), were moved to Petri dishes and incubated within the zebrafish embryo E3 medium (5 mM sodium chloride, 0.17 mM potassium chloride, 0.33 mM calcium chloride, and 0.33 mM magnesium sulfate) at a temperature of 28 °C. To evaluate in vivo toxicity and possibly developmental defects caused by CHX gluconate and EO water, the zebrafish embryos were individually exposed to E3 culture medium (negative control), 0.2% and 2.0% CHX gluconate (positive control), and EO waters comprising 0.0125% and 0.0250%, for 0.5 and 1.0 min using a yellow 100 µm cell strainer (Falcon^®^, One Becton Circle, Durham, NC, USA), then, transferred to Petri dishes (*n* = 10), and three repetitions and recorded data at representative stages (24, 48, and 72 hpf). The mortality and conditions of the embryos were captured under a light microscope (Olympus SZX16, Shinjuku-ku, Tokyo, Japan) and a digital camera (Canon EOS 550D, Ohta-ku, Tokyo, Japan) under 40× and 100× magnifications. Percentage mortality rate of the zebrafish embryos and the body length were recorded and calculated.

### 2.3. Antimicrobial Efficacy

Prior to the test, *Streptococcus mutans* (*S. mutans,* ATCC^®^ 25175) cultures were inoculated on the surface of tryptic soya agar with polysorbate 80, lecithin (Sigma-Aldrich/51414, St. Louis, MO, USA), and 5% defibrinated sheep blood at 37 °C, for 48–72 h, and then the microbial count was adjusted to about 10^8^ colony-forming units (cfu)/mL. One hundred microliters microbial counts of 1 × 10^8^ cfu/mL *S. mutans* was used to mix with each 10 mL specimen of 0.2% and 2.0% CHX gluconate (positive control) and EO water (containing 0.0125% and 0.0250% HOCl) to give an inoculum of 10^5^ to 10^6^ cfu/mL for designed contact time periods. A suitable amount of sample was collected immediately from each suspension and the value of cfu/mL in each suspension was determined by plate-count method according to United States Pharmacopeia (USP) Chapter 51 antimicrobial effectiveness testing.

### 2.4. Statistical Analysis

An online web statistical calculator for one-way analysis of variance (ANOVA) with post-hoc Tukey HSD test (https://astatsa.com/OneWay_Anova_with_TukeyHSD/) was used to evaluate the statistical significance of the measured data. The Tukey HSD (“honestly significant difference”) post-hoc test was used to determine the significance of deviations (*p* < 0.05) in the measured data of each group.

## 3. Results

### 3.1. Preparation of Electrolyzed Oxidizing (EO) Waters

The pH value and the ORP for EO waters comprising 0.018% hypohalous acid produced from various salts are summarized in Table 1. Unlike the typical EO water electrolyzer equipped with a permeation membrane to individually produce acidic EO water and alkaline EO water, the bottle-type electrolyzer produced an EO water mixture from the product streams of anode and cathode. The harvested EO waters were alkaline, thus additional 0.1N hydrochloric acid solution was used to adjust the pH value to a pH of 5.5–6.5. The higher the pH value, the less the ORP of alkaline EO water.

To elucidate the pH dependence of the ORP, the EO water prepared from sodium chloride was chosen as the model system. Figure 1 is the pH dependence of the ORP for the EO water comprised of 0.003%, 0.0125%, and 0.0250% HOCl. The maximum ORP level of hypochlorous acid (HOCl) is between pH 2 to 3, and it decreases when the pH is below 2 and above 3. The ORP of EO water decreases with the concentration of HOCl when the pH is over 3, however, it is not a linear response.

### 3.2. In Vivo Toxicity Test

#### 3.2.1. Screening Test for EO Waters Prepared from Different Salts

Figure 2 and Figure 3 show the representative record chart and the photomicrographs for the hatch rate and mortality rate of zebrafish embryos exposed to different EO waters containing 0.018% hypohalous acid, for a soaking time of 3.0 min, at specific time periods. The percent mortality rate and the body length at 72 hpf of zebrafish embryos after exposure to EO water produced from various salts are summarized in Table 2. Zebrafish embryos were hatched and grew healthily under condition of the E3 medium (negative control) up to 72 hpf. The EO water prepared from the sodium chloride showed a 0% mortality rate similar to the zebrafish embryos in E3 medium, but mortality rates of 53.3 ± 33.3% and 100.0 ± 0.0% were observed for the EO waters prepared from sodium bromide and calcium chloride, respectively.

#### 3.2.2. In Vivo Toxicity Test for Various EO Waters and Chlorhexidine (CHX) Gluconate Mouthwashes

Figure 4 and Figure 5 show the representative record chart and the photomicrographs for the hatch rate and mortality rate of zebrafish embryos soaking in different EO waters, CHX gluconate, and control group of E3 medium for 0.5 min. All the zebrafish embryos were dead in 2.0% CHX gluconate after 0.5 min soaking time while zebrafish embryos hatched and bred healthily in E3 medium and EO waters (0.0125% and 0.0250% HOCl) up to 72 hpf.

The soaking time dependence on the mortality rates of zebrafish embryos exposed to various concentrations of HOCl or CHX gluconate were investigated. Table 3 shows the mortality rates and body lengths at 72 hpf for zebrafish embryos after exposure to 0.0125% HOCl, 0.0250% HOCl, and E3 medium. Except for the percent mortality rate for the group of 0.0250% HOCl with 1.0 min and E3 medium, there were no significant differences among the rest of the groups (*p* > 0.05).

### 3.3. Antimicrobial Properties for EO waters and CHX Gluconates

The bacterial counts and reduction of *S. mutans* before and after treatment are listed in Table 4. All the HOCl containing EO waters or 2.0% CHX gluconate treatment groups showed over a 5 log 10 cfu/mL reduction in *S. mutans* population, indicating >99.9% antimicrobial efficacy.

## 4. Discussion

Electrolyzed oxidizing (EO) water has been widely used as a disinfectant in agriculture, dentistry, medicine, and the food industry in recent years, due to its effective disinfection, easy operation, and environmental safety [28]. Production of EO water needs only water and salt. The main reactions of EO water production in an electrolytic cell are shown in Table 5 [18].

In the anode chamber, halogen gas like chlorine can react with water to form HOCl and HCl. In addition, the HOCl dissociates into the hypochlorite ion (OCl^−^) and the hydrogen ion (H^+^), depending on pH value. The typical maximum level of HOCl is between a pH of 4 and 5.5 and it decreases when the pH is below 4 and above 5.5 [29]. However, our experimental results showed that the maximum ORP level of HOCl was between pH 2 and 3. S. Wei et al. reported similar results for EO water with low pH (2.2–2.7) and high ORP (>1100 mV) produced by electrolysis of a dilute NaCl solution in an electrolysis chamber with a Pt/Ti electrode [30]. The electrode material is the key parameter related to current efficiency and stability for the formation of active chlorine in EO water production [31].

The storage stability of EO water is an important issue that needs to be overcome before its possible us in medical applications. The storage stability of EO water bottles made of amber glass and high-density polyethylene (HDPE) were investigated by measuring the dependence of pH, oxidation-reduction potential (ORP), and free chlorine concentration with storage conditions through various time periods [32]. The HOCl decomposition was determined as first-order decay, and the decay rate of the stored samples in the HDPE bottle was faster than that in an amber glass bottle. The storage stability of EO water under the influence of different light, agitation, and packaging conditions was examined. The effect of diffused light was more significant as compared with other factors [33].

Our results showed that the 0.2% CHX gluconate group revealed a bacterial reduction rate of 87.5% and 97.1% for treatment times of 0.5 min and 1.0 min, respectively. The EO waters comprised of 0.0125% and 0.0250% HOCl both showed >99.9% antimicrobial efficacy but with zero mortality rate under in vivo toxicity test. However, the mortality rate was 66.7 ± 26.2% in 0.2% CHX gluconate for a soaking time of 0.5 min. In a word, high antimicrobial efficacy but low toxicity was observed for EO waters with 0.050% and 0.0250% HOCl as compared with 0.2% CHX gluconate.

Chlorhexidine (CHX) (CAS 55-56-1) is an effective antiseptic with a low water solubility of 0.08 g/100 mL (20 °C) due to its symmetrical structure comprised of four chlorophenyl rings and two biguanide groups linked by a central hexamethylene bridge [34]. The water solubility of CHX digluconate (CAS 18472-51-0) is over 50% (*w*/*v*), possibly through micelles formation caused by gluconate. Continuing rinsing with 0.2% CHX gluconate mouthwash has been reported to be effective but revealed some side effects in decreasing the saltiness of sodium chloride and the bitterness of quinine [35]. To deal with this dilemma, a low concentration of 0.12% CHX digluconate was proposed for its possibly fewer side effects, while holding similar efficacy in controlling dental plaque and gingivitis [36]. A double-masked randomized clinical study to compare the CHX concentration effect on plaque, bleeding, and side effects was reported by M. Haydari et al. [37], however, with opposite conclusions, such as the fact that 0.2% CHX was reported to have considerably better plaque inhibiting effects than 0.12% CHX and 0.06% CHX. Optimal rinsing time is an important factor for a mouthwash when reflecting a real usage situation and product testing conditions. A rinsing time of 30 s appeared to be sufficient for all plaque-covered surfaces for intraoral distribution (spread) of mouthwashes [38]. No significant difference was observed in the level of plaque after 72 h of non-brushing, whether or not the subjects rinsed for 15, 30, or 60 s with 0.2% CHX gluconate [39].

HOCl, known as a reactive oxygen species (ROS), can be naturally synthesized using human immune system cells [40], or by an electrochemical technique [41]. It has been demonstrated to be a wide antimicrobial spectrum for the inhibition of multiple microorganisms with important anti-inflammatory and proliferative activity but without side effects such as irritation of the mucosa [42,43]. All of the above advantages make EO water comprising HOCl a good candidate in the development of a mouthwash for dental caries and periodontal disease control.

An extensive investigation on the antimicrobial properties of mouthwash comprising 0.050% and 0.0250% HOCl was carried out by D. M. Castillo et al. [26]. The effect on *P. gingivalis* and *S. mutans* was determined by SDS-PAGE. CHX showed a higher efficiency than HOCl against *S. mutans*, *A. israelii*, *E. corrodens,* and *E. cloacae*, while HOCl was more effective than CHX against *P. gingivalis*, *A. actinomycetemcomitans*, *C. rectus,* and *K. oxytoca*. Concerning the subjective user experience with mouthwashes containing 0.025% and 0.050% HOCl, a randomized controlled trial was conducted [25]. At the clinical examination 24 h after rinsing, no clinical or mucosal changes or dental pigmentation were observed. However, it was claimed that the HOCl containing mouthwashes had the most unpleasant taste and dry tissue sensation among the evaluated products. In fact, the reported minimum bactericidal concentration (MBC) values of HOCl for different test organisms was in the range of 5.6–12.5 ppm for killing bacteria in less than 1 min [42]. Further detailed evaluations about its clinical effective and user acceptance are needed regarding EO water with 0.0125% HOCl as a potential mouthwash.

## 5. Conclusions

Within the limitations of this study, we concluded that both EO waters comprising 0.0125% and 0.0250% HOCl revealed remarkable, but similar, antimicrobial properties (>99.9%) to that of conventional 2.0% CHX gluconate against *S. mutans*. The EO waters both showed similar mortality rates to E3 medium but little in vivo toxicity, revealing the possibility of their application as an alternative mouthwash for oral hygiene and dental care. An additional limitation of this study was the transferability from animals to humans.

## Figures and Tables

**Figure 1 materials-13-04299-f001:**
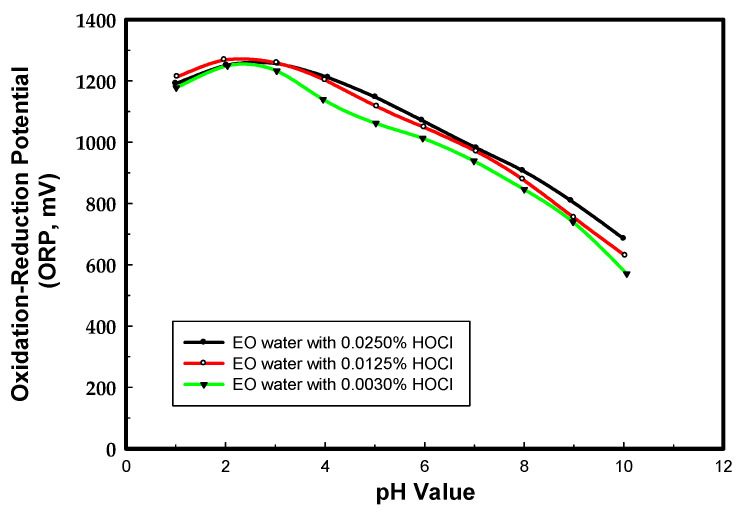
The pH dependence of the ORP for EO waters comprising various HOCl contents.

**Figure 2 materials-13-04299-f002:**
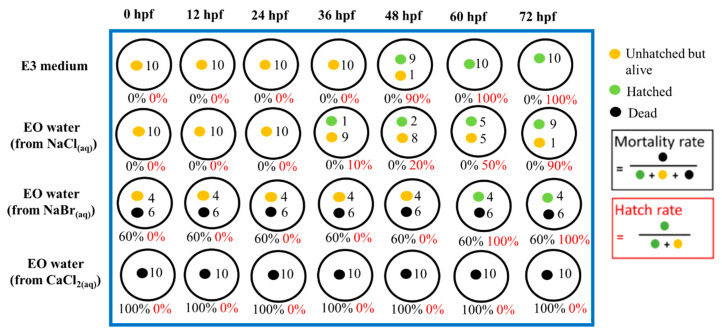
The hatch rate and mortality rate of zebrafish embryos in E3 medium (negative control) and various EO waters with 0.018% hypohalous acid produced from various salts at different time periods, up to 72 hph.

**Figure 3 materials-13-04299-f003:**
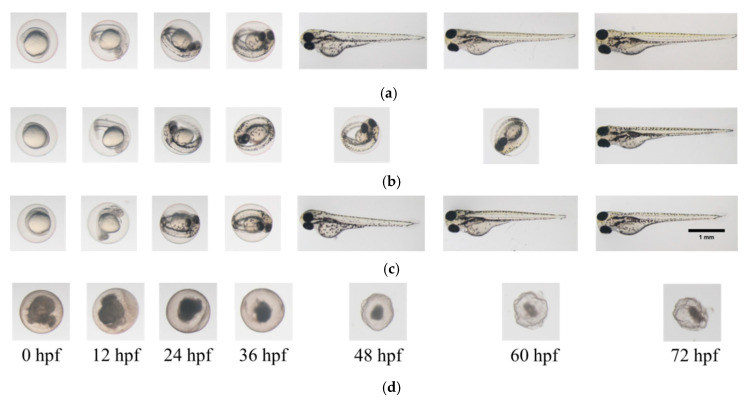
Photomicrographs of zebrafish embryos. (**a**) In E3 medium (negative control); (**b**) In EO water prepared from NaCl with 0.018% HOCl; (**c**) In EO water prepared from NaBr with 0.018% HOBr; (**d**) In EO water prepared from CaCl_2_ with 0.018% HOCl for soaking time of 3 min.

**Figure 4 materials-13-04299-f004:**
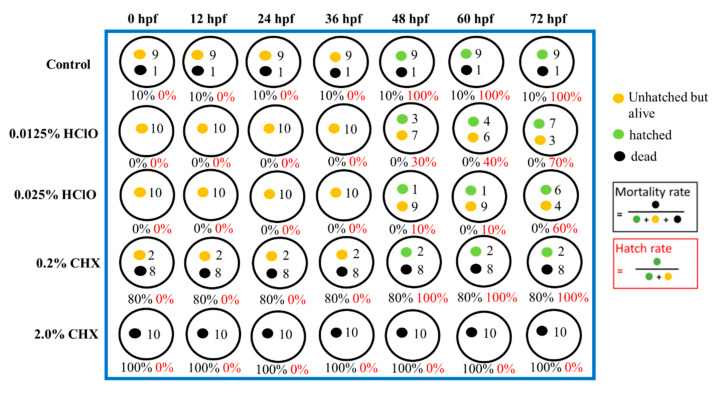
The hatch rate and mortality rate of zebrafish embryos in E3 medium (negative control), 0.0125% HOCl, 0.0250% HOCl, 0.2% chlorhexidine (CHX) gluconate, and 2.0% CHX gluconate for soaking of 0.5 min.

**Figure 5 materials-13-04299-f005:**
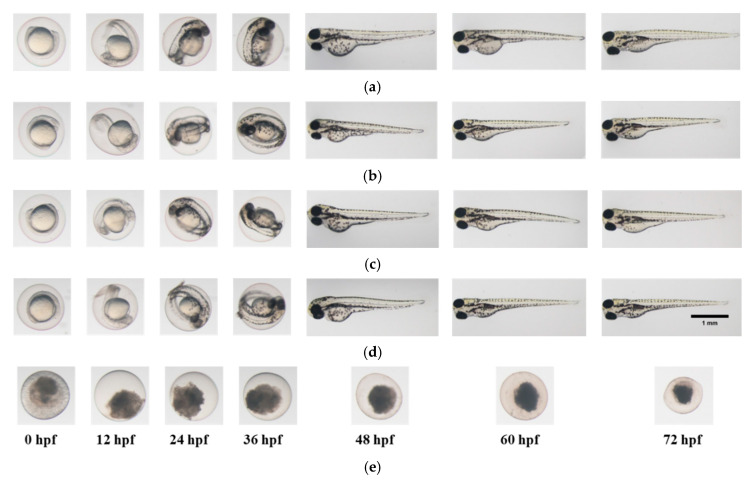
Photomicrographs of zebrafish embryos in (**a**) E3 medium; (**b**) 0.0125% HOCl; (**c**) 0.0250% HOCl; (**d**) 0.2% CHX gluconate and (**e**) 2.0% CHX gluconate, for a soaking time of 0.5 min.

**Table 1 materials-13-04299-t001:** The pH dependence of the oxidation reduction potential (ORP) for electrolyzed oxidizing (EO) waters comprising various concentrations of hypochlorous acid (HOCl).

Samples	pH	ORP (mV)
EO water prepared from NaCl (0.018% HOCl)	8.69	772
EO water prepared from NaBr (0.018% HOBr)	10.26	700
EO water prepared from CaCl_2_ (0.018% HOCl)	7.26	945

**Table 2 materials-13-04299-t002:** Percent mortality rate and the zebrafish body length at 72 hpf after 3 min soaking time in EO waters produced from various salts.

Samples	3 min Soaking Time	
	Mortality Rate * (%)	Body Length ** (mm)
E3 Medium	0.0 ± 0.0 ^a^	3.62 ± 0.14 ^d^
EO water prepared from NaCl (0.018% HOCl)	0.0 ± 0.0 ^a^	3.50 ± 0.27 ^d^
EO water prepared from NaBr (0.018% HOBr)	53.3 ± 33.3 ^b^	3.58 ± 0.05 ^d^
EO water prepared from CaCl_2_ (0.018% HOCl)	100.0 ± 0.0 ^c^	N.A.

* Mortality rate (*N* = 3 and each well loading 10 zebrafish embryos); ** zebrafish body length (*N* = 30). Values are shown as the mean ± standard deviation. Mean values followed by the same superscript letter do not significantly differ (*p* > 0.05) according to post-hoc test. N.A.: not available.

**Table 3 materials-13-04299-t003:** Percent mortality rates and the body lengths at 72 hpf of zebrafish embryos after exposure to E3 medium, HOCl, or CHX gluconate.

Samples	0.5 min Soaking Time	
	Mortality Rate * (%)	Body Length ** (mm)
E3 Medium	3.3 ± 4.7 ^a^	3.53 ± 0.12 ^d,f^
EO water comprising 0.0125% HOCl	0.0 ± 0.0 ^a^	3.46 ± 0.02 ^e^
EO water comprising 0.0250% HOCl	0.0 ± 0.0 ^a^	3.45 ± 0.07 ^e^
0.2% CHX gluconate	66.7 ± 26.2 ^b^	3.43 ± 0.12 ^e,f^
2.0% CHX gluconate	100.0 ± 0.0 ^c^	N.A.

* Mortality rate (*N* = 3 and each well loading 10 zebrafish embryos); ** body length (*N* = 30). Values are the mean ± standard deviation. Mean values followed by the same superscript letter do not significantly differ (*p* > 0.05) according to post-hoc test. N.A., not available.

**Table 4 materials-13-04299-t004:** Bacterial counts and reduction of *S. mutans* (ATCC^®^ 25175) before and after treatment.

Samples	Treatment Time (min)	Before (cfu/mL)	After (cfu/mL)	Bacterial Reduction (%)
EO water comprising 0.0125% HOCl	0.5	4.3 × 10^5^	<1	>99.9
EO water comprising 0.0250% HOCl	1.0	4.3 × 10^5^	<1	>99.9
0.2% CHX gluconate	0.5	4.3 × 10^5^	5.3 × 10^4^	87.5
0.2% CHX gluconate	1.0	4.3 × 10^5^	1.3 × 10^4^	97.1
2.0% CHX gluconate	0.5	4.3 × 10^5^	4.5 × 10^0^	>99.9
2.0% CHX gluconate	1.0	4.3 × 10^5^	<1	>99.9

**Table 5 materials-13-04299-t005:** The main reactions of EO water production from different salts.

Salts	Anode	Cathode
	H_2_O → 2H^+^ + 1/2 O_2_↑ + 2e^−^	H_2_O + e^−^ → OH^−^ + 1/2 H_2_↑
NaCl	2NaCl → Cl_2_↑+ 2e^−^ + 2Na^+^	Na^+^ + OH^−^ → NaOH
	Cl_2_↑+ H_2_O → HOCl + H^+^ + Cl^−^	
NaBr	2NaBr → Br_2_↑ + 2e^−^ + 2Na^+^	Na^+^ + OH^−^ → NaOH
	Br_2_↑+ H_2_O → HOBr + H^+^ + Br^−^	
CaCl_2_	CaCl_2_ → Cl_2_↑ + 2e^−^ + Ca^+2^	Ca^+2^ + 2OH^−^ → Ca(OH)_2_
	Cl_2_↑+ H_2_O → HOCl + H^+^ + Cl^−^	

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
