# Peer review of "The In Vivo Toxicity and Antimicrobial Properties for Electrolyzed Oxidizing (EO) Water-Based Mouthwashes"

_materials, 2020, doi:10.3390/ma13194299_

Round 1
Reviewer 1 Report
The title of this manuscript does not fully describe the work carried out. The title implies that chlorhexidine is part of the study, whilst to my understanding the novelty of these investigations lies on the testing of electrolysed oxidising water as compared with chlorhexidine as the gold standard the title should better reflect the work The title also mentions mouthwashes, but it doesn't seem that mouthwashes were tested.
The abstract does not introduce nor justify the study. Some clear justification is required in an abstract.
The methods are not very well described. One big ommision is the preparation of chlorhexidine (normally obtained as a powder) - the manuscript does not include details of the diluent of chlorhexidine, nor how the EO waters and chlorhexidine were diluted to achieve the different concentrations tested. The microbial testing is not well explained. Again the bacteria tested were grown on the surface of an agar plate, but normally disinfectant testing is carried out in a liquid environment, overall the methods for the antimicrobial testing do not include sufficient detail to enable another researcher to replicate the experiments carried out. The controls for the toxicity testing are well described but in my opinion not appropriate unless both the chlorhexidine and the EO waters were diluted in the E3 medium, but this is not explained so it is impossible to determine. For the microbiological testing, there are not evident controls carried out - as the counts of treated samples are compared to the counts prior to the test. There is also no evidence of use of a neutraliser necessary for this type of experiment (such as BS EN 1276:2009).
The results are not sufficiently clearly presented - maybe in some kind of graph rather than tables. In particular, the data for antimicrobial testing, the overall bacterial reduction is presented as a percentage, when for this kind of test a log reduction (in line with standardised disinfectant testing methodologies) is more appropriate.
The discussion does not fully tackle the results obtained nor it fully describes its relevance. Again, the structure of the discussion needs improvement as it lacks in flow which would aid comprehension.
Reviewer 2 Report
Dear authors,
I appreciate the efforts of the manuscript with the title “The in Vivo Toxicity and Antimicrobial Properties for Chlorhexidine (CHX) Gluconate and Electrolyzed Oxidizing (EO) Water-Based Mouthwashes”.
Introduction: hypotheses is missing
Please check all table and figure numbers throughout the manuscript. Text citations of the tables in the results section seem to be in disorder.
What were the individual sample sizes, was a sample size calculated before the evaluation?
Statistics: add information of the software, was the data normally distributed?
Please explain all abbreviation the first time they are mentioned
Add limitations of this study, such as the transferability from an animal to humans
Round 2
Reviewer 1 Report
Following the first round of revisions, I still don't feel that my comments are fully addressed. I still have particular issue with the lack of detail regarding controls for both the toxicological studies and the antimicrobial studies - as these still are not fully described in the text and without a full description of controls this manuscript is not suitable for publication.
The other major issue I raised regarding the title has been tackled in a satisfactory way. Equally, I also commented on the lack of detail of the diluents of the compounds under test, there is now some text clarifying to this effect.
Reviewer 2 Report
Accept
Taking into account the changes made to the text during the revision, the publication is now acceptable.
